# Mast Cells Retard Tumor Growth in Ovarian Cancer: Insights from a Mouse Model

**DOI:** 10.3390/cancers15174278

**Published:** 2023-08-26

**Authors:** Nicole Meyer, Nicole Hinz, Anne Schumacher, Christine Weißenborn, Beate Fink, Mario Bauer, Sophie von Lenthe, Atanas Ignatov, Stefan Fest, Ana Claudia Zenclussen

**Affiliations:** 1Experimental Obstetrics and Gynecology, Medical Faculty, Otto-von-Guericke University Magdeburg, 39108 Magdeburg, Germany; nicole.meyer@ufz.de (N.M.); anne.schumacher@ufz.de (A.S.); atanas.ignatov@med.ovgu.de (A.I.); stefan.fest@ufz.de (S.F.); 2Department of Environmental Immunology, UFZ-Helmholtz Centre for Environmental Research Leipzig-Halle, 04318 Leipzig, Germany; beate.fink@ufz.de (B.F.); mario.bauer@ufz.de (M.B.); 3Perinatal Immunology, Saxonian Incubator for Clinical Translation (SIKT), Medical Faculty, Leipzig University, 04103 Leipzig, Germany; 4Department of Pediatrics, Städtisches Klinikum Dessau, Academic Hospital of University Brandenburg, 06847 Dessau-Rosslau, Germany

**Keywords:** mast cells, ovarian carcinoma, mouse model, mast cell-deficiency, ultrasonography

## Abstract

**Simple Summary:**

Approximately 225,000 new cases of ovarian cancer are diagnosed annually worldwide, of which 140,000 die. The aim of the study was to assess the role of mast cells in ovarian cancer pathology. We confirmed a suppressive effect of mast cells on ovarian tumor growth both in vitro and in vivo. An accurate understanding of the interaction between immune defenses and malignant cells is of enormous importance for improving current treatments and establishing new ones against ovarian cancer.

**Abstract:**

Ovarian cancer has the highest mortality rate among female reproductive tract malignancies. A complex network, including the interaction between tumor and immune cells, regulates the tumor microenvironment, survival, and growth. The role of mast cells (MCs) in ovarian tumor pathophysiology is poorly understood. We aimed to understand the effect of MCs on tumor cell migration and growth using in vitro and in vivo approaches. Wound healing assays using human tumor cell lines (SK-OV-3, OVCAR-3) and human MCs (HMC-1) were conducted. Murine ID8 tumor cells were injected into C57BL6/J wildtype (WT) and MC-deficient C57BL/6-Kit^W-sh/W-sh^ (Kit^W-sh^) mice. Reconstitution of Kit^W-sh^ was performed by the transfer of WT bone marrow-derived MCs (BMMCs). Tumor development was recorded by high-frequency ultrasonography. In vitro, we observed a diminished migration of human ovarian tumor cells upon direct or indirect MC contact. In vivo, application of ID8 cells into Kit^W-sh^ mice resulted in significantly increased tumor growth compared to C57BL6/J mice. Injection of BMMCs into Kit^W-sh^ mice reconstituted MCs and restored tumor growth. Our data show that MCs have a suppressive effect on ovarian tumor growth and may serve as a new therapeutic target.

## 1. Introduction

Fully functional ovaries are the prerequisite for maternal contribution to the emergence of new life. Here, follicles mature into oocytes that get fertilized in the fallopian tubes while traveling to the uterus, where they implant as blastocysts in the maternal endometrium and develop into an embryo. However, the ovaries can also be the origin of death, as all the cells that make up the ovaries can develop into different types of cancer. When detected at a very early stage, 70% of patients survive for more than 10 years after diagnosis. Unfortunately, only about 25–30% of ovarian cancers are found at an early stage, as the symptoms become more obvious when the disease has spread [1]. Consequently, ovarian cancer has the highest mortality rate among female reproductive tract malignancies, with a 5-year survival rate of only 45% [2]. Based on the type and stage of cancer, local treatments, including surgery and radiation, or systemic treatments, including chemotherapy, hormone therapy, and targeted drug therapy, are used alone or in combination. However, these treatments cannot overcome the high mortality rate. Thus, there is an urgent need for a coherent understanding of the cellular key players and underlying mechanisms of tumor development to improve existing therapies and develop new strategies to save the lives of women all over the world. 

Mast cells (MCs) are innate immune cells that are best studied for their participation in allergic immune reactions. However, we are just beginning to understand the multiple functions of MCs that arise from their plasticity and pleiotropism [3]. For instance, they accumulate in a wide range of human and animal tumors and their microenvironments [4]. One of the unique properties of MCs is the accumulation of numerous mediator-filled cytoplasmic granules. The variety and different nature of these mediators are the reason for the versatile functions of the MCs. They are able to secrete pro-angiogenic and growth factors in addition to pro- and anti-inflammatory mediators [3]. Thus, MCs possess both pro-tumorigenic and anti-tumorigenic capabilities, leading to progression or arrest of tumor growth, depending on the type of tumor. The participation of MCs in ovarian tumor pathophysiology is poorly understood. 

The main aim of our study was to understand the participation of MCs in ovarian cancer. To fulfill our aim, we first performed experiments to unravel MC effects on in vitro tumor growth. Secondly, studies employing a mouse model where ovarian carcinoma cells were injected into MC-deficient mice and reconstituted with bone marrow-derived MCs (BMMCs) were carried out to understand the in vivo contribution of MCs to tumor establishment and growth. A profound understanding of the role of MCs in ovarian cancer development and progression is of critical importance for the development of new targeted therapies.

## 2. Materials and Methods

### 2.1. Cell Lines

The human ovarian cancer cell lines SK-OV-3 and OVCAR-3 (Cell Lines Services, Eppelheim, Germany) were cultured in DMEM/F12 (PAN Biotech, Aidenbach, Germany) supplemented with 10% fetal bovine serum (FBS) and 100 U/mL penicillin, 100 μg/μL streptomycin (P/S). The human MC line HMC-1 (kindly provided by Dr. J. H. Butterfield, MN, USA) was maintained in IMDM medium (Gibco, Schwerte, Germany) supplemented with 10% FBS and 1% P/S. The mouse ovarian cancer cell line ID8 [5] was maintained in DMEM (Gibco, Schwerte, Germany) supplemented with 4% FBS, 1% L-Glutamine, 1% P/S, and 1% insulin-transferrin-selenium (Gibco, Schwerte, Germany). 

All cell lines were cultured at 37 °C in a humidified 5% CO_2_ atmosphere. The medium was replaced every 2–3 days, and cells were detached with 0.05% trypsin-EDTA (Life Technologies, Darmstadt, Germany) when they reached 90% confluence.

### 2.2. Wound Healing Assay

A total of 1.5 × 10^5^ ovarian cancer cells per well were plated in 12-well plates. After 16 h, scratches were generated with a 100 µL pipette tip. Ovarian cancer cells were co-cultured with HMC-1 (cell-to-cell ratio 1:1 or 1:5) directly or by using 0.4 µm pore-size transwells (Corning, NY, USA). Ovarian cancer cells alone served as controls. Images of scratch areas were taken after 0 h, 24 h, and 48 h with the light microscope and Axio Vision software version 4 (Zeiss, Jena, Germany).

### 2.3. Biological Specimens

Human ovarian tumor samples (n = 11) were collected with written informed consent from the patients. The study was performed in accordance with the principles of the Declaration of Helsinki. The use of the samples was approved by the Ethics Committee of the Otto-von-Guericke University of Magdeburg (2808, March 2014). Tissues were employed for immunohistochemistry. 

### 2.4. Toluidine Blue Staining

In order to visualize MCs, human ovarian tumor sample sections (5 µm thickness) were stained with toluidine blue (Carl Roth, Karlsruhe, Germany). After standard dewaxing and rehydration procedures, slides were incubated for 45 s in a 0.1% toluidine blue solution and rinsed briefly in distilled water. Slides were dehydrated by dipping them 10 times in ethanol (75%, 95%, and 100%) and incubating them in xylene twice for 2 min each. Afterwards, slides were covered with Roti-Histokitt (Carl Roth, Karlsruhe, Germany).

### 2.5. Mice

This study was carried out in accordance with the recommendations of the Ministry of Saxony-Anhalt, Germany. The protocol was approved by the “Landesverwaltungsamt Sachsen Anhalt: 42502-2-1296UniMD.” Mice were housed in our barrier facility with a 12 h light/dark cycle and received food and water ad libitum. 

B6.Cg-KitW-sh/HNihrJaeBsmJ mice (C57BL/6J background; Kit^W-sh^) and their WT controls, C57BL/6J, were bred in our facility. 

### 2.6. Tumor Injection

Following pilot experiments to establish optimal conditions (data not included in the paper), 5 × 10^6^ ID8 cells were injected subcutaneously into the flanks of 8–10-week-old female C57BL6/J mice or MC-deficient Kit^W-sh^ mice. Control mice received 0.1 mL of PBS. Tumor development was recorded weekly by high-frequency ultrasonography. The mice were sacrificed 14 weeks after tumor application (day 98). 

### 2.7. MC Reconstitution in Kit^W-sh^ Mice with BMMCs

Isolation and maturation of WT BMMCs were performed as described elsewhere [6]. On the two following days, the first being recorded as day 0 and the start of the experiment, 5 × 10^5^ BMMCs were injected into the tail vein of Kit^W-sh^ mice (systemic reconstitution). After allowing and confirming reconstitution of MCs in a subset of animals, 12 weeks later, tumor injection (as described before) took place. Tumor growth was recorded weekly by high-frequency ultrasonography. The mice were sacrificed 14 weeks after tumor application. 

### 2.8. High-Frequency Ultrasonography

The Vevo^®^ 2100 system (Fujifilm VisualSonics, Amsterdam, Netherlands, transducer MS550D-0421) was used for the ultrasound measurements [7]. The hair on the flank was removed by using depilatory cream (Reckitt Benckiser, Hull, UK). Pre-warmed ultrasound gel (Gello GmbH Geltechnik, Ahaus, Germany) was applied to the depilated skin. B-Mode was used to visualize anatomical structures in a 2D grayscale image. Tumor length and width were measured, and tumor volume (mm^3^) was calculated using the following formula: $$V=\pi \;\times \;\frac{3}{4}\;\times \;\frac{length}{2}\;\times \;\frac{width}{2}\;\times \;\frac{depth}{2}$$

Ultrasound examinations were performed every two weeks (days 0, 14, 28, 42, 56, 70, 84, and 98). The data were analyzed with VisualSonics VEVO LAB 5.6.1 software. 

### 2.9. RNA Isolation, cDNA Synthesis and Real-Time qPCR

RNA isolation: Total RNA was isolated using TRIzol^®^ reagent according to the manufacturer’s instructions. Briefly, frozen tissues were homogenized in TRIzol with an Ultra Turrax T25 homogenizer (NeoLab, Heidelberg, Germany). RNA was extracted with chloroform, precipitated with isopropanol, washed with ethanol, and diluted in RNAse-free water. RNA purity and quantity were determined by measuring the UV absorbance at 260 and 280 nm. RNA samples were stored at −80 °C.

cDNA synthesis: RNA (2 µg) was incubated with oligo dT primers and RNAse-free water for 10 min at 75 °C. After 2 min on ice, the marked mRNA was incubated with dNTPs (2.5 mmol/mL), DNAse (2 U/µL), and RNAse inhibitor (40 U/µL) in a reaction buffer for 30 min at 37 °C. DNAse was inactivated for 5 min at 75 °C, and the samples were incubated on ice for 2 min. Reverse transcriptase and RNAse inhibitor were added, and cDNA was synthesized at 42 °C for 60 min. Following this, reverse transcriptase was inactivated at 94 °C for 5 min, and cDNA samples were stored at −20 °C.

Real-time polymerase chain reaction (RT-qPCR): The qPCR was performed on the Biomark HD system (Fluidigm, San Francisco, CA, USA) using BioMark™ 96.96 Dynamic Array chips according to the manufacturer’s recommendations. Intron-spanning primers were designed, and universal probe-library probes (UPL) were selected by the Universal Probe Library Assay Design Center (http://qpcr.probefinder.com/organism.jsp, accessed on 20 July 2021, currently no longer available, Appendix A). PCR was performed with FastStart Universal Probe Master Mix (Roche, Mannheim, Germany). All reactions were performed in duplicate. Gene expression values were normalized to the mean of the four reference genes *Actb*, *Gapd*, *Rplp0*, and *Ubc* and additionally normalized to the lowest measured value over all samples per gene. This leads to relative gene expression on the log2 scale (relative unit). 

### 2.10. Statistical Analysis

The normality of the data sets was assessed with the D’Agostino–Pearson omnibus test before analyzing the data with either parametric or non-parametric tests. Data are presented as medians or means, depending on their distribution, with S.D. The number of samples or mice, the statistical tests, and the *p* values for the experiments are indicated in the figure legends. All results were confirmed in at least three independent experiments. Statistical analyses were performed with GraphPad Prism 9.0.

## 3. Results

### 3.1. MCs Are Present in Human Ovarian Cancer Samples

To investigate whether MCs are present in human ovarian tumors and surrounding tissue, we performed toluidine blue staining of human ovarian tumor samples (n = 11) and found MCs to be present within the cancer tissue. Representative pictures of four patients are shown in Figure 1. 

### 3.2. MCs Reduce the In Vitro Migration of Human Ovarian Cancer Cells

To better understand whether the presence of MCs within tumor tissue may indicate their involvement in tumor establishment, maintenance, and/or growth, we next co-cultured human ovarian cancer cells (SK-OV-3, OVCAR-3) with a human MC line (HMC-1) in different cell-to-cell ratios (1:1 and 1:5) with or without transwells. By performing a wound healing assay, we observed that the addition of MCs to the culture diminished the migration of SK-OV-3 (Figure 2A) and OVCAR-3 (Figure 2B) cells. For OVCAR-3 cells, this effect was observed for both cultures with and without the use of transwells. This means that both direct cell-to-cell contact and MC-secreted soluble factors may be responsible for MC-dependent growth reduction. For SK-OV-3 cells, the effect was observed exclusively with the use of transwells, pointing to an inhibitory effect of soluble factors on tumor growth. 

### 3.3. Optimization of an In Vivo Model to Study the Participation of MCs in Ovarian Cancer Growth

Next, we injected mouse ovarian cancer cells subcutaneously into the flanks of female C57BL6/J WT mice in order to establish an in vivo ovarian cancer model. The tumor development was recorded by high frequency ultrasonography and tumor volume increased over time (Figure 3A). A representative image of the tumor measurements is shown in Figure 3B. In control mice that received PBS tumor growth did not occur. Toluidine blue staining of tumor specimens confirmed the presence of MCs at the tumor site (Figure 3C).

Our C57BL/6J-based model shows a tumor growth rate comparable to the one reported for other published models [8] and is further adequate to study the participation of MCs in tumor growth as these are present at the tumor site in wild-type mice.

### 3.4. MC Absence Enhances In Vivo Ovarian Carcinoma Tumor Growth

To unequivocally confirm the role of MCs in inhibiting ovarian carcinoma, we inoculated MC-deficient Kit^W-sh^ mice with ovarian cancer cells as described above. Impressively, tumor growth was significantly augmented in MC-deficient mice already at early time points (day 28) compared with MC-sufficient controls (Figure 4A). Representative ultrasound images and tumor photographs are shown in Figure 4B. 

### 3.5. Reconstitution of MC-Deficient Mice with BMMCs Diminishes Tumor Growth

To unequivocally confirm that inhibition of tumor growth in Kit^W-sh^ mice was due to the absence of MCs, we next reconstituted them with bone marrow-derived mast cells (BMMCs). BMMCs obtained from C57BL/6 WT mice (purity of the population shown in Appendix A) reconstituted the MC pool. After reconstituting Kit^W-sh^ mice with BMMCs, we observed that the tumor growth was significantly slower than in the MC-deficient Kit^W-sh^ mice and comparable with that in MC-sufficient WT animals (Figure 5A). Representative ultrasound images are shown in Figure 5B.

### 3.6. Unraveling Changes in the Tumor Environment Driven by the Absence or Presence of MCs in the Host

We next performed a comparative RT-qPCR analysis of marker expression among the ID8-injected C57BL/6J, ID8-injected Kit^W-sh^, and BMMC + ID8-injected Kit^W-sh^ mice. Target panels include cytokines, chemokines, MC-related markers, and markers associated with tumor behavior, such as invasion. Markers were changed in MC-deficient Kit^W-sh^ compared to MC-sufficient WT mice, and BMMC treatment restored a profile similar to that of control WT mice (Figure 6 and Appendix A). In particular, *Il3* was decreased and *Il23* was increased in the tissue of MC-deficient animals compared to controls and normalized upon BMMC reconstitution. MC-related molecules were, as expected, diminished or absent in MC-deficient mice and normalized upon reconstitution. Accelerated tumor growth in MC-deficient mice was accompanied by significant inhibition of *Ifna1*, which was normalized upon BMMC reconstitution.

## 4. Discussion

Ovarian cancer is one of the most common gynecological cancers and ranks first in mortality among other cancers of the female reproductive tract [9]. Approximately 70% of patients diagnosed with ovarian cancer will experience a recurrence [10]. This high recurrence rate is a serious threat to women´s health worldwide. As classical therapies do not succeed in eradicating the tumor, the identification of immune cells and understanding of their functions within the tumor microenvironment are critical to understanding tumor escape mechanisms during cancer progression.

MCs are pleiotropic, highly plastic innate immune cells that store a variety of mediator-filled granules in their cytoplasm. Their functions and effects are versatile, including physiological and pathophysiological roles. Although MC infiltration has been reported in a wide range of human and animal tumors, the significance of MCs remains uncertain [4]. It is known that MCs are able to modulate the tumor microenvironment through pro-tumor actions (angiogenesis induction, extracellular matrix degradation, and direct and indirect immune suppression) on the one hand and anti-tumor actions (direct tumor cell cytotoxicity and immune cell recruitment and activation) on the other hand [11]. As described by Jammal et al., and confirmed here, MCs are present within tumor tissue from patients with OC [12]. However, the participation of MCs in ovarian tumor pathophysiology is grossly underinvestigated and therefore poorly understood. 

In the present study, we aimed to understand the impact of MCs on ovarian tumor cell migration and tumor growth using in vitro and in vivo approaches. The OVCAR-3 cell line, a human high-grade serous ovarian adenocarcinoma cell line, was co-cultured with the human MC line HMC-1. We observed a diminished migration of ovarian tumor cells upon direct and indirect, soluble mediator-based contact with MCs. By contrast, HMC-1 diminished the migration of SK-OV-3 ovarian cancer cells exclusively through the use of transwells, which means that only indirect but not direct cell-to-cell contact leads to inhibition of tumor growth. Nonetheless, our in vitro results suggest that MCs can have a direct suppressive influence on ovarian tumor cell migration, and this scenario is physiologically and clinically relevant as MCs are detectable in human ovarian cancer specimens. In previous studies, it was shown that recruitment and activation of MCs in tumors are mainly mediated by tumor-derived stem cell factor (SCF) that interacts with the c-kit receptor on MCs [13]. The membrane-bound form of SCF is found in high levels in ovarian cancer cells [14].

The use of MC-deficient mouse models like Kit^W-sh^ has significantly increased our knowledge of MC biology and function. Kit^W-sh^ mice are characterized by profound MC-deficiency in all tissues but normal levels of major classes of other differentiated hematopoietic and lymphoid cells [15]. A comparison of results between WT and MC-deficient mice provides an opportunity to confirm the MC specificity of the observed effects in MC-repopulated tissues upon reconstitution of the mice with BMMCs. In vivo application of ovarian tumor cells into MC-deficient Kit^W-sh^ mice in our study resulted in significantly increased tumor growth when compared to C57BL6/J MC-sufficient mice. Thus, we demonstrated that the absence of host MCs is related to abnormally rapid tumor growth in ovarian cells. We suggest that MCs have a hampering effect on ovarian tumor growth. Reconstitution of MC-deficient mice with BMMCs diminished tumor growth compared to MC-deficient mice, and the tumor kinetic growth and tumor size at the end of the experiment were both comparable to the wild-type controls. These results unequivocally confirm that MCs are capable of inhibiting in vivo ovarian carcinoma tumor growth. 

In line with our study, different studies show that MC infiltration in tumor areas is associated with a higher survival rate in patients. Chan and colleagues observed that not only the number of MCs but also vessel density are important predictors for the survival rate. They found that peri-tumoral MC infiltration in tumors with high mean vessel density predicts improved survival in women with advanced (stage III–IV) epithelial ovarian cancer [16]. Another study from 2009 showed a correlation between tumor-infiltrating tryptase-positive MCs and improved clinical outcome in patients with malignant pleural mesothelioma [17]. In other cancer types, MC infiltration correlated with favorable tumor characteristics and improved patient survival in prostate cancer [18] and breast cancer [19].

In contrast to these studies that position MCs as rather tumor-fighting cells, other studies show that the presence of MCs in the tumor microenvironment is associated with cancer growth or a poor prognosis in ovarian cancer. Liu and colleagues demonstrated the histamine-induced proliferation of ovarian cancer cells. Histamine promoted the proliferation of ovarian cancer cells by upregulating the expression of estrogen receptors α and β [20]. MCs and basophils are the most relevant sources of histamine in the immune system [21]. Analyzing the tumor microenvironment of ovarian cancer cells showed more activated MCs in the early stages compared with the advanced stages of the disease. MC localization was mainly in the stroma [22]. The abundance of stromal tumor-infiltrating MCs predicted a poor prognosis for high-grade serous ovarian cancer, characterized by increased infiltration of pro-tumor cells and impaired anti-tumor immune functions [23]. Further, higher frequencies of MCs are reported to be present in malignant ovarian neoplasms compared to benign ovarian neoplasms [12]. A risk score model predicting the prognosis of ovarian cancer patients was proposed by Sun and colleagues, in which associations between mRNA alternative splicing clusters and tumor environments were analyzed. Correlation analysis results showed that three types of immune cells, namely activated MCs, resting NK cells, and neutrophils, had a positive correlation with the risk score [24]. Another study showed that MC chymase (Mcpt5) and tryptase (Mcpt6 and 7) activity increased at all stages of tumor progression, whereas the number of MC remained constant [25]. A functional proteomics screen of proteases in colorectal carcinoma found MC proteases at very high levels in adjacent normal tissue and not detectable in the metastases. Whether MC-derived proteases serve a host-protective function or are involved in tumor progression is under investigation [26].

In our study, comprehensive qPCR analysis of tumor samples revealed significant differences between the experimental groups regarding the MC-associated markers Cpa3, Il-3 as well as the MC-proteases (Mcpt1, 2, 4, 5, 6, 7). All other markers, excluding Il-23 and Ifnα1, were comparable between the groups. This suggests that MCs might be responsible for the observed effects in terms of tumor growth. Many publications show that expression of MC proteases correlates with MC maturation, angiogenesis, and tumor progression. By contrast, other studies speculate that MCs may promote inflammation, inhibition of tumor cell growth, and tumor cell apoptosis by releasing mediators such as IL-1, IL-4, IL-6, IL-8, Mcpt3, Mcpt4, INF-y, TNF-α, TGF-β, LTB4, and chymase [27]. Tumor growth is mainly dependent on neovascularization and can be effectively inhibited by various anti-angiogenic agents. Interestingly, MCs are capable of secreting anti-angiogenic factors. Murata and colleagues identified prostaglandin D2 as a MC-derived anti-angiogenic factor in expanding solid lung carcinoma that governs the tumor microenvironment by restricting excessive responses to vascular permeability and TNF-α production [28]. Other studies showed that MC mediators have direct tumor cytotoxic [29] and tumor growth inhibitory [30] effects. Next to the direct effect of MCs and mediators on tumor growth, it is also possible that MCs modulate anti-tumor immunity by recruiting and activating other immune effector cells like dendritic cells, NK cells, CD8^+^ cytotoxic T lymphocytes, and CD4^+^ Th1 cells at tumor sites [11]. Whether this is true for our mouse model or for patients is not known and is worth including in future studies.

## 5. Conclusions

In conclusion, the discovery of new target cells is relevant for improving current treatments and establishing new treatments and immunotherapies against ovarian cancer. MCs are unique candidates for combined treatment modalities due to their radiation resistance on the one hand and, on the other, their localization in close proximity to blood vessels, which makes them sensitive to immunotherapies. Our study positions them as putative candidates to be further studied as therapy targets, as we could show that they are able to inhibit tumor growth both in vitro and in vivo.

## Figures and Tables

**Figure 1 cancers-15-04278-f001:**
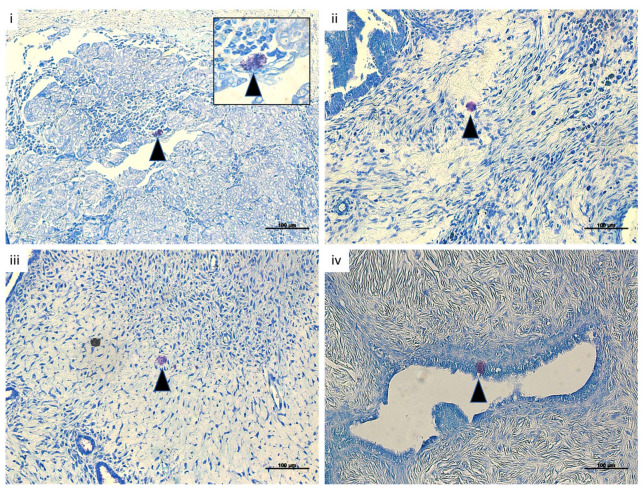
MCs are present in human ovarian cancer samples in vivo. Representative toluidine blue-stained sections of four (**i**–**iv**) human ovarian cancer samples (n = 11) showing MCs (arrow) within the tumor tissue (magnification 20×, scale bar = 100 µm). Square shows higher magnification (magnification 100×).

**Figure 2 cancers-15-04278-f002:**
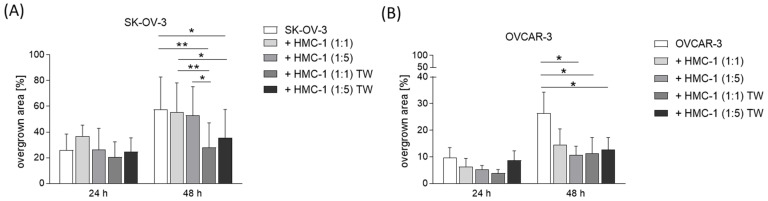
MCs reduce the migration of human ovarian cancer cells in vitro. Overgrown area of SK-OV-3 (**A**) or OVCAR-3 (**B**) cells co-cultured with HMC-1 cells in different cell-to-cell ratios (1:1, 1:5) with or without transwell at 24 h and 48 h (n = 3). Results are presented as mean with SD. Statistical analysis was performed using a two-way ANOVA followed by Bonferroni’s multiple comparison test (* *p* < 0.05, ** *p* < 0.01).

**Figure 3 cancers-15-04278-f003:**
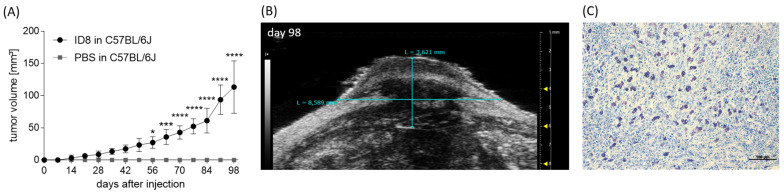
MC presence in an ID8 cell-derived tumor. (**A**) Tumor volumes of ID8-derived tumors in C57BL/6J mice (n = 7) and PBS-treated C57BL/6J mice (n = 3) from day 0 until day 98 after s.c. ID8 cell injection. (**B**) Representative grey-scale images of in vivo tumor measurements (length, width) via ultrasonography in B-Mode at day 98. (**C**) Representative toluidine blue-stained section of an ID8 tumor showing MCs (purple) within the tumor tissue (scale bar = 100 µm). Results are presented as means with SD. Statistical analysis was performed using a two-way ANOVA followed by Bonferroni´s multiple comparison test (* *p* < 0.05, *** *p* < 0.001, **** *p* < 0.0001).

**Figure 4 cancers-15-04278-f004:**
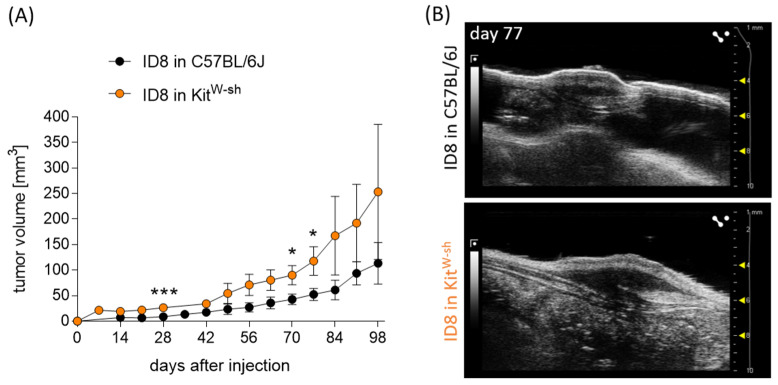
Significantly increased tumor growth in MC-deficient Kit^W-sh^ mice compared to WT. (**A**) Tumor volume in C57BL/6J (n = 7) and Kit^W-sh^ mice (n = 5) from day 0 until day 98 after tumor cell injection. (**B**) Representative grey-scale images of the ID8-derived tumors at day 77 in C57BL/6J and Kit^W-sh^ mice. Results are presented as means with SD. Statistical analysis was performed using a two-way ANOVA followed by Bonferroni´s multiple comparison test (* *p* < 0.05, *** *p* < 0.001).

**Figure 5 cancers-15-04278-f005:**
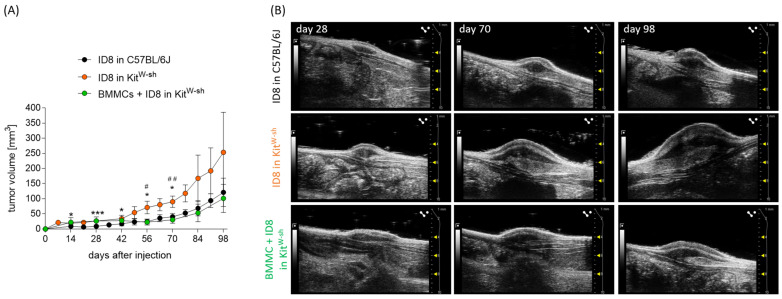
Diminished tumor growth in BMMC-reconstituted Kit^W-sh^ mice. (**A**) Tumor volume in C57BL/6J (n = 12), Kit^W-sh^ (n = 5), and BMMC-reconstituted Kit^W-sh^ (n = 5) mice from day 0 until day 98 after tumor cell injection. (**B**) Representative grey-scale images of ID8-derived tumors at days 28, 70, and 98 in C57BL/6J, Kit^W-sh^, and BMMC-treated Kit^W-sh^ mice. Results are presented as means with SD. Statistical analysis was performed using a two-way ANOVA followed by Bonferroni´s multiple comparison test (*^, #^
*p* < 0.05, ^##^
*p* < 0.01, *** *p* < 0.001; *, comparison of ID8 in C57BL/6J and ID8 in Kit^W-sh^; #, comparison of ID8 in C57BL/6J and BMMC + ID8 in Kit^W-sh^).

**Figure 6 cancers-15-04278-f006:**
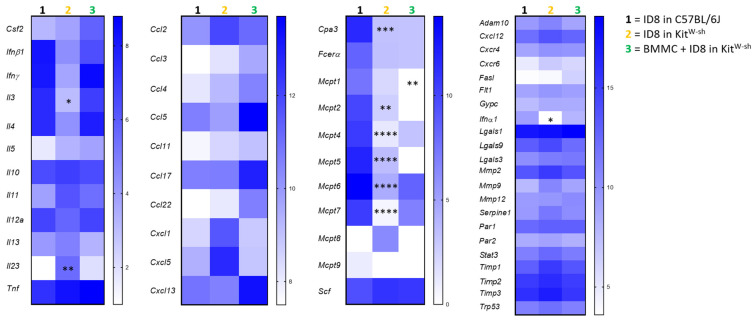
Absence or presence of MCs affects the tumor environment. Visualization of molecules analyzed with RT-PCR of tumor tissue of ID8-injected C57BL/6J (n = 9) and Kit^W-sh^ (n = 4), and BMMC-reconstituted Kit^W-sh^ mice (n = 3). Results are presented as heatmap showing medians. Statistical analysis was performed using a two-way ANOVA followed by Bonferroni´s multiple comparison test (* *p* < 0.05, ** *p* < 0.01, *** *p* < 0.001, **** *p* < 0.0001).

## Data Availability

The datasets generated during and/or analyzed during the current study are available from the corresponding author on reasonable request.

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
