# Peer review of "Mast Cells Retard Tumor Growth in Ovarian Cancer: Insights from a Mouse Model"

_cancers, 2023, doi:10.3390/cancers15174278_

Round 1

Reviewer 1 Report

Meyer et al. demonstrated that mast cells have a suppressive effect on ovarian tumor growth and may serve as a new therapeutic target. The paper is very interesting, but I have several issues I need to address.

1. Wound healing images for the OVCAR-3 cell line are not suitable, and the cells are randomly distributed. I recommend replacing these figures with other suitable figures. I also recommend performing a transwell assay.

2. What about the effect of MSc on the apoptosis and cell cycle arrest of human ovarian cancer cells (SK-OV-3, OVCAR-3(?

3. PCR results should be presented in a table.

4. A schematic diagram summarizing the conclusion should be provided.

5. There are minor corrections, such as lines 23 and 232; the manuscript should be revised carefully.

Reviewer 2 Report

This manuscript investigates mast cells (MCs) in ovarian cancer, focusing on their impact on tumor growth. Through in vitro co-culture experiments, the authors observe reduced migration of ovarian cancer cells in the presence of MCs. In vivo tests using MC-deficient mice (KitW-sh) reveal heightened tumor growth, while reconstitution with bone marrow-derived MCs (BMMCs) reinstates growth inhibition. The study also investigates gene expression changes, highlighting MC-related markers and cytokine alterations in tumors. The findings suggest that MCs inhibit ovarian tumor growth and propose them as potential therapeutic targets, prompting further exploration. However, key questions remain:

1. Have the authors investigated the impact of MCs on cell growth in co-culture experiments? Additionally, can they explore the molecular mechanisms underlying MCs' inhibitory effects on tumor cell migration or growth, focusing on key signaling pathways or mediators in MC-cancer cell interactions?

2. Could there be crosstalk between MCs and other immune cells in the tumor microenvironment? Do MCs modulate the activity of immune cells like T cells, dendritic cells, or macrophages in human/mouse tumor samples?

3. Considering MC targeting as a therapeutic strategy, could the authors examine the effects of modulating MCs on tumor growth in animal models using MC-specific inhibitors/activators? Can mast cell-focused strategies be integrated into existing ovarian cancer treatments, including chemotherapy or immunotherapy?

4. Is there potential clinical relevance of MC presence in human ovarian tumor samples, correlated with patient survival or treatment response?

5. Could a comprehensive transcriptomic analysis be conducted using high-throughput techniques to identify specific genes and pathways influenced by MC presence or absence, offering a deeper understanding of MC effects on the tumor microenvironment?

Round 2

Reviewer 1 Report

The authors covered all my concerns. I recommend the acceptance of the paper.

Reviewer 2 Report

The authors have address all my concerns and I have no further questions